# *ImageScope*: Unifying Language-Guided Image Retrieval via Large Multimodal Model Collective Reasoning

## Abstract

With the proliferation of images in online content, language-guided image retrieval (LGIR) has emerged as a research hotspot over the past decade, encompassing a variety of subtasks with diverse input forms. While the development of large multimodal models (LMMs) has significantly facilitated these tasks, existing approaches often address them in isolation, requiring the construction of separate systems for each task. This not only increases system complexity and maintenance costs, but also exacerbates challenges stemming from language ambiguity and complex image content, making it difficult for retrieval systems to provide accurate and reliable results. To this end, we propose *ImageScope*, a training-free, three-stage framework that leverages collective reasoning to unify LGIR tasks. The key insight behind the unification lies in the compositional nature of language, which transforms diverse LGIR tasks into a generalized text-to-image retrieval process, along with the reasoning of LMMs serving as a universal verification to refine the results. To be specific, in the first stage, we improve the robustness of the framework by synthesizing search intents across varying levels of semantic granularity using chain-of-thought (CoT) reasoning. In the second and third stages, we then reflect on retrieval results by verifying predicate propositions locally, and performing pairwise evaluations globally. Experiments conducted on six LGIR datasets demonstrate that *ImageScope* outperforms competitive baselines. Comprehensive evaluations and ablation studies further confirm the effectiveness of our design.

## CCS Concepts

• **Information systems → Information retrieval**; **Retrieval models and ranking**; **Users and interactive retrieval**.

## Keywords

Language-Guided Image Retrieval, Large Multimodal Model, Collective Reasoning

**ACM Reference Format:**
Anonymous Author(s). 2018. *ImageScope*: Unifying Language-Guided Image Retrieval via Large Multimodal Model Collective Reasoning. In *Proceedings of Make sure to enter the correct conference title from your rights confirmation emai (Conference acronym 'XX)*. ACM, New York, NY, USA, 16 pages. https://doi.org/XXXXXXX.XXXXXXX

## 1 Introduction

The past decade have witnessed an explosion of multimodal information on the internet, particularly with images emerging as one of the most prevalent mediums for online information sharing. Numerous image-centric platforms have proliferated, such as Instagram, Flickr, and Pinterest. To extract valuable information from the vast amount of images available on the web, image retrieval [15, 33] has evolved into a rapidly developing technology that underpins various applications in real life, especially in fields like e-commerce [11, 69] and search engines [61]. The traditional content-based image retrieval [24, 45, 57] and tag-based image retrieval [19, 54, 60] have achieved remarkable efforts, laying the foundation for the widespread adoption of text-to-image retrieval (TIR) [10] in most modern search engines. In recent years, TIR has been greatly boosted with Vision-Language Models (VLMs) [39, 40, 49, 53, 66] based on Transformer [55], which aligns visual and linguistic modalities within a joint latent space through pre-training on large-scale image-text pairs, providing advancements in retrieval accuracy and relevance.

Although these steady progress has been made, TIR falls short in capturing user's search intent in an interactive manner. Consequently, new tasks such as Composed Image Retrieval (CIR) [6, 13, 27, 46, 56] and Chat-based Image Retrieval (Chat-IR) [17, 37, 38] have been introduced. To be more specific, CIR enables users to refine search results through language feedback based on a provided reference image. As illustrated in Figure 1 (b), a user may wish to modify specific visual elements (*e.g.*, objects, attributes, and environments) of the given reference image, and she/he can provide language feedback to guide the system in retrieving images that align with the desired changes. In contrast, Chat-IR, as depicted in Figure 1 (c), focuses on progressively narrowing down the search results through multiple rounds of dialog interaction, especially when the user's retrieval intent is initially vague or evolves throughout the retrieval process. For instance, a user might start with a broad query like "A man walking on the street" and later specify a preference for visual elements such as "rainy day" or "cityscape" after reviewing initial results. Both CIR and Chat-IR allow for continuous refinement of results to accommodate the dynamic nature of user needs. Tasks like these, including text-to-image retrieval (TIR), all rely on user-provided textual input, and these tasks are generally termed as *Language-Guided Image Retrieval* (LGIR) [13, 25, 28]. The research on LGIR has evolved rapidly, making remarkable progress across various tasks [7, 20, 26, 35, 37, 38].

Despite these task-specific advances, a fundamental challenge remains: existing methods tend to address each task in isolation, focusing on optimizing for specific input modalities or interaction styles without providing a unified framework that generalizes across LGIR tasks. This fragmented approach limits the ability to integrate information from diverse inputs, such as combining reference images and multi-turn dialog, which is critical for handling

**Figure 1: Illustration of three language-guided image retrieval tasks: text-to-image retrieval, composed image retrieval and chat-based image retrieval.**

ambiguous queries and enhancing user's search experience. Moreover, the inherent ambiguity of natural language, combined with the complexity of real-world image content, makes it difficult to fully capture user intent and refine retrieval results. Accurately identifying subtle visual details remains particularly challenging for current methods, and the issue could be further amplified when user feedback is incomplete or imprecise.

To achieve this goal, in this paper, we propose a unified three-stage framework, named *ImageScope*, for LGIR, leveraging the advantages of multimodal collective reasoning to fully harness the potential of Large Multimodal Models (LMMs). The general idea underlying the unification is grounded in the compositional nature of language, allowing for the conversion of diverse LGIR tasks into a standardized text-to-image retrieval process. Moreover, the reasoning capacities of LMMs act as a universal means of verification to improve the precision of results. To establish a unified framework, we utilize LMM to generate textual descriptions for both input reference image and images in database. We set the semantic composition in the language domain, using Large Language Model (LLM) to synthesize the user's various forms of textual feedback into a coherent description of the target image. Then the retrieval is transformed into a text-to-image retrieval process, which can be executed by a pretrained VLM. Subsequently, a carefully designed reflective assessment incorporating a verification-evaluation paradigm is introduced to enhance the refinement of the results.

More specifically, the *ImageScope* framework consists of three stages. (1) **Stage 1: Semantic Synthesis.** To thoroughly analyze operations on visual elements referenced in the textual feedback, we define five distinct instruction types within a carefully tailored prompt: addition, removal, modification, comparison and retention. The LLM-based *reasoner* utilizes chain-of-thought (CoT) reasoning to integrate these operations, generating target image descriptions at three levels of granularity: core elements, enhanced details, and full synthesis, to address potential ambiguities in the user's feedback. Following this, a pre-trained VLM conducts dual-path retrieval for both text-to-image and text-to-text tasks to ensure robustness. (2) **Stage 2: Predicate Verification.** To overcome the limitations of pre-trained VLMs in capturing fine details, we propose a local semantic validation method based on predicate logic. The *reasoner*, guided by carefully crafted prompts, generates a series of verifiable propositions derived from the operations in the first stage. An LMM is then employed as a *verifier* to check the candidate images against these propositions. Additionally, we introduce a relaxation strategy to quantify the number of satisfied propositions, using this count to prioritize and rank the candidate

images. (3) **Stage 3: Overall Evaluation.** In this stage, we perform a holistic evaluation to determine whether the retrieved images fully meet the user's instructions, particularly in scenarios involving comparisons with a reference image. Another LMM, serving as an *evaluator*, is employed to iteratively narrow down the candidate images through pairwise comparisons, until the image that best satisfies the user's requirements is identified.

In our method, these multimodal models collaborate across different stages of reasoning, a cohesive three-stage framework. Additionally, the proposed *ImageScope* framework is highly flexible, and seamlessly compatible with various models without additional training. The outputs from each stage are user-friendly and offer a degree of interpretability.

To sum up, our main contributions are threefold:

- This paper presents a novel framework, *ImageScope*, designed to address language-guided image retrieval (LGIR) tasks. To the best of our knowledge, *ImageScope* is the first unified framework capable of handling various LGIR tasks without requiring additional training.

- We propose a reflection method called verification-evaluation for image retrieval task that accounts for both local and global semantics. This method combines predicate proposition with pairwise comparison, significantly improving retrieval performance.

- The experimental results on six prevalent LGIR datasets show that our framework achieves state-of-the-art performance. Ablation studies and in-depth analysis further validate the effectiveness and generality of *ImageScope*.

## 2 Related Work

### 2.1 Language-Guided Image Retrieval

Unlike traditional content-based [24, 45, 57] or tag-based image [19, 54, 60] retrieval methods, language-guided image retrieval (LGIR) encompasses a range of language-centric tasks, such as text-to-image retrieval, composed image retrieval (CIR), and chat-based image retrieval (Chat-IR), offering a retrieval paradigm that allows flexible language feedback. Early traditional CIR methods treat textual instructions as modifications to a reference image [7, 13, 46, 56], relying heavily on expensive annotated triplets for training data. Zero-shot CIR [6, 51] has been recently introduced to alleviate such reliance, which can be broadly classified into two categories: text inversion [41, 51] and LLM editing [34, 63]. In contrast, Chat-IR originally stemmed from visual dialogue [17] and visual question-answering (VQA) tasks [4, 22], where multiple rounds of

conversation revolve around a specific image to answer visual questions [18, 48]. Recent studies have designed a questioner to ask more discriminative questions [37, 38], aiding in better retrieval, and used LLM-based approaches to combine semantics for retrieval. However, these studies tend to address each LGIR task independently, lacking a unified modeling. In contrast, our framework adopts a training-free method to handle LGIR tasks in a unified manner, which significantly distinguishes it from previous approaches.

## 2.2 Large Models and Reasoning

In recent years, large language models (LLMs) [2, 50, 62, 67] and large multimodal models (LMMs) [5, 14, 43] have demonstrated remarkable capabilities across various tasks, particularly in generation, understanding, and planning. Researchers have found that step-by-step reasoning [58] and in-context learning can significantly enhance the performance of LLMs. Some studies have explored the impact of different reasoning structures on performance, such as chain [12], tree [64], and graph [8] structures. Additionally, given the known susceptibility of LLMs to hallucinations [30], some research attempts to mitigate errors in the reasoning process through validation mechanisms, either via the model's own feedback [47] or external feedback [21]. By contrast, another line of research focuses on decomposing complex problems for more effective solutions. The Least-to-Most [70] approach breaks problems down top-down into subproblems, while QDMR [29] decomposes them into directed acyclic graphs. These studies further promote advancements in areas like external tool usage [52] and multimodal question answering tasks [68]. Our work differs from these studies by designing a general reflection mechanism for LGIR tasks, which leverages the reasoning capabilities of LLMs and LMMs to refine retrieval results and enhance accuracy.

## 3 Methodology

In this section, we first formalize the LGIR task (§3.1), followed by an explanation of the unification approach to LGIR tasks as illustrated in Figure 2 (§3.2). Finally, we elaborate each stage of our proposed framework (§3.3, §3.4, §3.5).

## 3.1 Problem Definition

Let us define the image database $\mathcal{D}$, which consists of a set of images $\{I_i\}_{i=1}^N$. The goal of LGIR is to establish a scoring function $\mathcal{S} = \Psi(\mathcal{T}, I_r, \mathcal{D})$, where $\mathcal{T}$ represents the input text, $I_r$ denotes the input reference image, and $\mathcal{S}$ denotes the corresponding image scores. Then the images can be ranked according to their scores to produce the retrieval results. Based on this, **text-to-image retrieval** can be defined as $\mathcal{S} = \Psi(\mathcal{T}_{\text{desc}}, \emptyset, \mathcal{D})$, where $\mathcal{T}_{\text{desc}}$ represents the text description and $\emptyset$ indicates no reference image input. Similarly, given a reference image $I_r$ and a textual instruction $\mathcal{T}_{\text{inst}}$, **composed image retrieval** can be expressed as $\mathcal{S} = \Psi(\mathcal{T}_{\text{inst}}, I_r, \mathcal{D})$. Furthermore, given a conversation history $\mathcal{T}_{\text{dial}} = \{d_1, d_2, \dots\}$, **chat-based image retrieval** can be represented as $\mathcal{S} = \Psi(\mathcal{T}_{\text{diag}}, \emptyset, \mathcal{D})$.

## 3.2 Unified Framework

Achieving a unified framework for LGIR is inherently difficult due to the diverse nature of modalities and input types, each with its own unique semantic structures. Bridging these differences to enable coherent image retrieval requires advanced reasoning across multiple input forms. To address these complexities, in *ImageScope*, we use a language-centric semantic synthesis approach. The core insight behind this framework is the compositional nature of language—leveraging language descriptions to combine semantics from various input types and modalities. Recent advancements in LLMs, particularly in content understanding and reasoning, offer a promising foundation for semantic composition within the language space. This motivates us to translate visual content into language descriptions. To bridge vision with language, we employ an LMM as a *captioner* to convert visual inputs into textual descriptions. Simultaneously, a pre-trained VLM transforms both images from the image database $\mathcal{D}$ and their corresponding textual descriptions into vector representations.

$$T_1, \dots, T_N = \text{Captioner}_{\text{LMM}}(\mathcal{D}), \quad (1)$$

$$\boldsymbol{V}_T = v_{t1}, \dots, v_{tN} = \text{VLM}(T_1, \dots, T_N), \quad (2)$$

$$\boldsymbol{V}_I = v_{i1}, \dots, v_{iN} = \text{VLM}(I_1, \dots, I_N), \quad (3)$$

where $T_1, \dots, T_N$ are corresponding text description of images, $\boldsymbol{V}_T \in \mathbb{R}^{N \times d}$ and $\boldsymbol{V}_I \in \mathbb{R}^{N \times d}$ are vector representation of captions and images respectively, $d$ is the dimension decided by the VLM.

Following this, a *reasoner* based on an LLM synthesizes the semantics of different tasks within the language space, ultimately generating textual descriptions of the target image. Specifically,

- For TIR, we synthesize the semantics of the textual description with a blank image.
- For CIR, we synthesize the semantics of reference image description with textual instruction.
- For Chat-IR, we synthesize the semantics of previous round's image description with the current round's textual feedback.

In this way, *reasoner* generates textual descriptions for the desired target image, transforming the LGIR query into text-to-image retrieval. Then the query can be process by the pre-trained VLM.

## 3.3 Stage 1: Semantic Synthesis

Next, we delve into the details and elaborate on the three stages of the proposed *ImageScope* framework. As previously mentioned, the entire framework consists of three stages, each designed to address specific challenges in LGIR tasks: ambiguity in language feedback, local semantic validation, and overall evaluation. As illustrated in Figure 2 (a), a user's language feedback may exhibit ambiguity and uncertainty, potentially failing to fully capture all relevant visual elements, which could lead to misunderstandings. Moreover, a single textual description may involve multiple operations on visual elements. Therefore, effectively understanding and parsing user instructions is crucial in LGIR tasks. To address this challenge, we propose a semantic composition strategy based on chain-of-thought (CoT) reasoning in the first stage of our approach.

Specifically, we define five types of atomic instructions on visual elements (including objects and attributes), namely: addition, removal, modification, comparison, and retention. As shown in Figure 2 (a), the given textual instruction can be decomposed into one or a combination of these atomic instructions $O = \{o_i\}_{i=1}^M$. Based on this decomposition, we generate target image descriptions at different levels of semantic granularity:

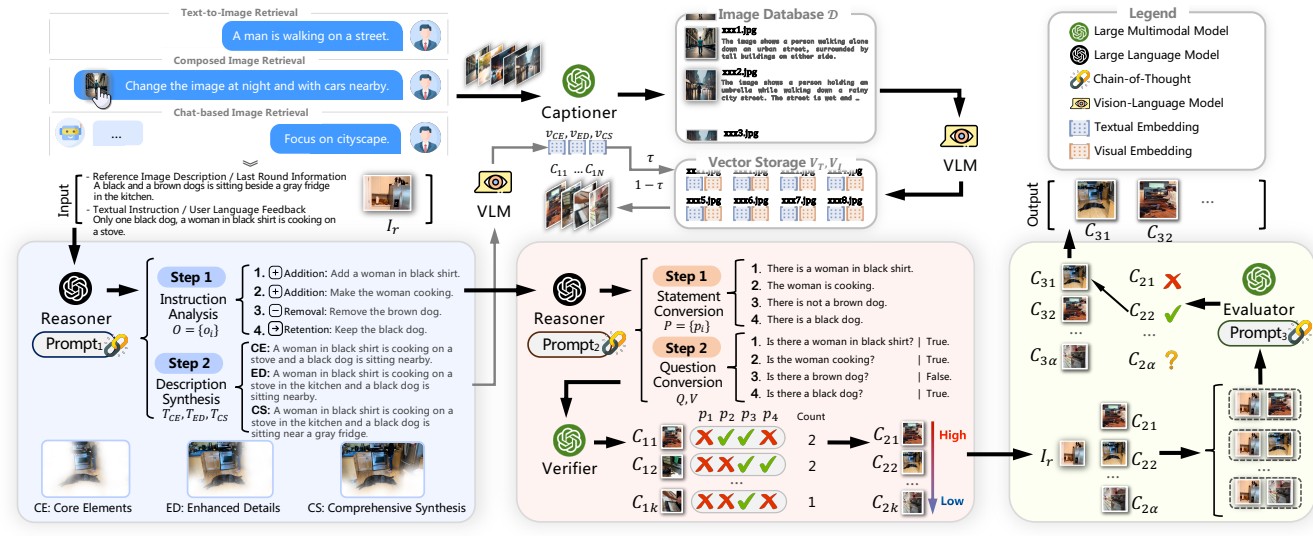

**(a) Stage 1: Semantic Synthesis**  **(b) Stage 2: Predicate Verification**  **(c) Stage 3: Overall Evaluation**

**Figure 2: Illustration of the proposed *ImageScope* framework.**

- Core Elements $T_{CE}$: Includes only the elements mentioned in the textual instruction.
- Enhanced Details $T_{ED}$: Includes elements from the textual instruction and necessary adjectives from the reference image.
- Comprehensive Synthesis $T_{CS}$: Includes the elements from textual instruction and relevant elements from reference image with necessary adjectives.

This process can be illustrated as:

$$T_r = \text{Captioner}_{\text{LLM}}(I_r) \tag{4}$$

$$O, T_{CE}, T_{ED}, T_{CS} = \text{Reasoner}_{\text{LLM}}(\mathcal{T}, T_r, \text{Prompt}_1), \tag{5}$$

where $T_r$ is the description of reference image $I_r$ and $\mathcal{T}$ is the input textual instruction. The Prompt$_1$ we use is shown in Figure 10. For TIR, we set $T_r$ as a blank image, and for Chat-IR, $T_r$ represents the last round information.

By synthesizing descriptions at multiple semantic granularities, we can more comprehensively capture the user's intent for the retrieval target. These descriptions are then encoded into embeddings through the text encoder of VLM. Both text-to-image retrieval and text-to-text retrieval are performed to enhance robustness. We introduce a parameter $\tau$ to control the weight between these two retrieval modes. The overall process is represented as follows:

$$v_{CE}, v_{ED}, v_{CS} = \text{VLM}(T_{CE}, T_{ED}, T_{CS}), \tag{6}$$

$$s = \frac{1}{3} \sum_{g \in \{CE, ED, CS\}} (\underbrace{\tau \cdot \text{sim}(v_g, \boldsymbol{V}_T)}_{\text{text-to-text}} + \underbrace{(1 - \tau) \cdot \text{sim}(v_g, \boldsymbol{V}_I)}_{\text{text-to-image}}), \tag{7}$$

where $s \in \mathbb{R}^{1 \times N}$ is the similarity scores vector of the query, and $\text{sim}(\cdot, \cdot)$ indicates cosine similarity. Finally, based on similarity scores, we obtain an initial ranking list of candidate images:

$$\{C_{11}, C_{12}, \ldots, C_{1N}\} = \text{argsort}_{\downarrow}(s), \tag{8}$$

where $\text{argsort}_{\downarrow}(\cdot)$ represents sorting in descending order based on the scores, $\{C_{11}, C_{12}, \ldots, C_{1N}\}$ denotes the image retrieval results of the first stage.

### 3.4 Stage 2: Predicate Verification

While the first stage typically yields relatively reliable results, certain retrieval outcomes may not accurately reflect user intent due to limitations in pre-trained VLMs in capturing nuanced details. Inspired by the reflection mechanisms in LLM reasoning, we propose a local semantic verification method based on predicate proposition to further refine the retrieval process, as depicted in Figure 2 (b). Leveraging the decomposed atomic instructions from the first stage, we employ a CoT strategy to guide reasoner in sequentially generating propositions $P = \{p_i\}_{i=1}^{M}$, question forms $Q = \{q_i\}_{i=1}^{M}$, and corresponding truth values $V = \{v_i\}_{i=1}^{M}$. The question form represents interrogative sentence, which can be answered by the *verifier* with a single Yes or No. The truth value represents the correct attribute reflected in the user's statement.

Building upon this foundation, the *verifier* addresses each candidate image by answering the question form $Q$ of proposition $P$. This process enables the determination of the correctness of each proposition. Ideally, candidate images meeting the retrieval criteria should satisfy conjunctive form $\bigwedge_{i=1}^{M} p_i \leftrightarrow v_i$ [1]. However, considering the performance limitations of the *verifier* and potential issues with images, requiring every proposition to be true may be overly stringent. Thus, we use a relaxation that allows for partial non-fulfillment of propositions, aiming to satisfy as many propositions as possible rather than demanding strict adherence to all. Specifically, for each candidate image $C_{1j}$, we calculate the number of propositions in the conjunctive form $\bigwedge_{i=1}^{M} p_i \leftrightarrow v_i$ that satisfy $(p_i \leftrightarrow v_i)$, denoted as $c_j = \text{Verifier}_{\text{LMM}}(\bigwedge_{i=1}^{M} p_i \leftrightarrow v_i, C_{1j})$, which is used to count the number of correct answers and where $c_j \in \mathbb{R}$ is the value for the $j$-th candidate image. Finally, candidate images are ranked according to the count value, where a higher count value indicates that the image better matches the user's retrieval intent. In the implementation, we use an LMM as the *verifier* to check the top-$k$ candidate images from the first stage $\{C_{11}, C_{12}, \ldots, C_{1N}\}$. During ranking, a stable sorting algorithm is employed to ensure

---

[1]$p_i \leftrightarrow v_i = (p_i \wedge v_i) \vee (\neg p_i \wedge \neg v_i)$, *i.e.*, $p_i$ and $v_i$ have the save value.

that images with higher similarity scores are prioritized when count values are equal. This process can be represented as follows:

$$P, Q, V = \text{Reasoner}_{\text{LLM}}(O, \text{Prompt}_2), \tag{9}$$

$$\boldsymbol{c} = \{c_j\}_{j=1}^k = \text{Verifier}_{\text{LMM}}\left(\bigwedge_{i=1}^M p_i \leftrightarrow v_i, C_{1j}\right) j = 1, \cdots, k, \tag{10}$$

$$\{C_{21}, C_{22}, \ldots, C_{2k}\} = \text{argsort}_{\downarrow}(\boldsymbol{c}), \tag{11}$$

where $O$ represents atomic instructions from the output of the first stage, $C_{1j}$ is the candidate image from the first stage, $\boldsymbol{c} \in \mathbb{R}^{1 \times k}$ denotes the count value vector. The $\text{Prompt}_2$ we use is shown in Figure 11. Then we can derive the refined retrieved images $\{C_{21}, \cdots, C_{2k}\}$ of the second stage.

### 3.5 Stage 3: Overall Evaluation

In the second stage, we focus on verifying the local semantics of the retrieved images. In contrast, the third stage involves an overall evaluation of candidate images, particularly in scenarios requiring comparison with reference images. To achieve this, we introduce an additional LMM as an *evaluator*. The *evaluator*'s task is to perform pairwise comparisons between the reference image and top-ranked candidate images from the second stage. By integrating image content and textual feedback, the *evaluator* determines whether the candidate images approximately meet the user's needs, providing binary results (Yes or No) along with necessary justifications. This process sequentially assesses each candidate until one meets the criteria or the threshold $\alpha$ is reached, which is the maximum number of images to evaluate. If a suitable candidate is found, it is re-ranked to the top. We also consider different forms of user feedback, *e.g.*, descriptions of desired changes or direct preferences for images, which are encoded into carefully designed prompt. This stage can be illustrated as:

$$\boldsymbol{f} = \{f\}_{i=1}^{\alpha} = \text{Evaluator}(I_r, C_{2j}, \text{Prompt}_3) \; j = 1, 2, \cdots, \alpha, \tag{12}$$

$$\{C_{31}, C_{32}, \cdots, C_{3\alpha}\} = \text{argsort}_{\downarrow}(\boldsymbol{f}), \tag{13}$$

where $\boldsymbol{f}$ is the binary results for candidate images, $\{C_{31}, C_{32}, \cdots, C_{3\alpha}\}$ represents the final ranking of images of the third stage. The $\text{Prompt}_3$ is shown in Figure 12.

By combining local verification with global evaluation, *ImageScope* leverages multimodal collective reasoning to ensure the top-ranked image satisfies user intents both in detail and overall.

## 4 Experiments

### 4.1 Experiment Setup

**Benchmark and Metrics.** We evaluate our framework for LGIR on six prevalent LGIR datasets. Specically, for CIR, we use CIRR [46], CIRCO [6] and FashionIQ [59]. CIRR is the first natural image dataset for CIR. It also designs a subset retrieval task with a group candidates from the image database. CIRCO expands the image database's scale and provides multiple ground truth annotations to mitigate false negative issue. FashionIQ focuses on fashion-domain, encompassing three categories: dress, shirt, and toptee. We adhere to the original benchmarks, employing Recall@k as the metric for CIRR and FashionIQ, and mean average precision (mAP@k) for CIRCO. For TIR, we use the widely adopted Flickr30K [65]

**Table 1: Benchmark details.**

| Dataset | Split | Type | # Queries | # Images |
|---|---|---|---|---|
| Flickr30K [65] | Test | TIR | 5,000 | 1,000 |
| MSCOCO [42] | Test | TIR | 25,010 | 5,000 |
| CIRR [46] | Test | CIR | 4,148 | 2,316 |
| CIRCO [6] | Test | CIR | 800 | 123,403 |
| FashionIQ-Shirt [59] | Val. | CIR | 2,038 | 6,346 |
| FashionIQ-Dress [59] | Val. | CIR | 2,017 | 3,817 |
| FashionIQ-Toptee [59] | Val. | CIR | 1,961 | 5,373 |
| VisDial [17] | Val. | Chat-IR | $2,064 \times 10$ | 50,000 |

and MSCOCO [42] datasets, both evaluated with Recall@k. For Chat-IR, we use VisDial [17] dataset and measure the multi-round performance with Hits@k [37, 38]. The details of these benchmarks are shown in Table 1.

**Baselines.** We compare *ImageScope* with various strong baseline methods. Given the training-free nature of *ImageScope*, our focus is primarily on zero-shot methods for a fair comparison. (1) For CIR, the baseline algorithms include PALAVRA [16], Pic2Word [51], SEARLE [6], iSEARLE [1], CIReVL [34], LDRE [63], HyCIR [32], LinCIR [23], and FIT4CIR [41]. (2) For TIG, we compare CLIP [49] and OpenCLIP [31] to demonstrate the performance improvement of the framework. (3) For Chat-IR, we evaluate against different versions of CLIP and PlugIR [37] method to assess its effectiveness.

**Implementation Details.** We implement our method using PyTorch [3], with vLLM [36] serving as the inference engine for both LLMs and MLLMs. The default models used for VLM, captioner, reasoner, verifier, and evaluator are CLIP-ViT-L/14 [31], LLaVA-v1.6-7B [44], LLaMA3-8B [2], PaliGemma-3B-mix-224 [9], and InternVL2-8B [14], respectively. Moreover, we further analyze the performance of different models in the discussion section. The temperature and top-p of sampling are set to 0 and 1 to ensure deterministic outputs. The weight $\tau$ in stage 1 is set to 0.15. The number of candidate images to verify in stage 2, *i.e.*, $k$ is set to 20. The number of images to evaluate in stage 3 $\alpha$ is set to 3. All experiments are conducted on a server equipped with A100-40G.

### 4.2 Performance Evaluation

**Composed Image Retrieval.** Table 2 presents the numerical results on CIRCO and CIRR test set, and average results of FashionIQ validation set. We group these methods based on different VLM configurations. As seen, it is evident that our *ImageScope* demonstrates remarkable performance across various CIR datasets. On CIRCO and CIRR datasets, it achieves state-of-the-art (SOTA) performance compared to numerous competitive methods. With CLIP-ViT-L/14 as the VLM backbone, *ImageScope* brings an absolute improvement of 5.01% on the mAP@5 metric for CIRCO, as well as absolute improvements of 12.84% and 15.93% on Recall@1 and Recall$_{\text{subset}}$@1 for CIRR, respectively, highlighting the framework's significant effectiveness. Regarding FashionIQ dataset, *ImageScope* still shows competitive performance compared to strong baselines, achieving the best or second-best metrics on the average result. Furthermore, from the table, we also have the following observations:

- The VLM remains the foundation for most methods. When scaling up the size of VLM from ViT-B/32 to ViT-L/14, almost all

Table 2: Performance comparison of CIR on CIRCO test set, CIRR test set and FashionIQ validation set. We report average results of three splits for FashionIQ. The best results are in boldface, and the second best results of baselines are underlined. "*" means using CLIP weights from [49]. "-" denotes results are not reported in the original papers. The complete experimental results are presented in Tables 4 and 5.

| VLM | Method | CIRCO | | | | CIRR | | | | | | | FashionIQ Avg. | |
| | | mAP@k | | | | Recall@k | | | | Recall_Subset@k | | | Recall@k | |
| | | k=5 | k=10 | k=25 | k=50 | k=1 | k=5 | k=10 | k=50 | k=1 | k=2 | k=3 | k=10 | k=50 |
|---|---|---|---|---|---|---|---|---|---|---|---|---|---|---|
| CLIP-ViT-B/32 | iSEARLE [1] | 10.58 | 11.24 | 12.51 | 13.26 | 25.23 | 55.69 | 68.05 | 90.82 | - | - | - | 24.40 | 44.80 |
| | iSEARLE-OTI [1] | 10.31 | 10.94 | 12.27 | 13.01 | 26.19 | 55.18 | 68.05 | 90.65 | - | - | - | 25.06 | 44.79 |
| | CIReVL [34] | 14.94 | 15.42 | 17.00 | 17.82 | 23.94 | 52.51 | 66.00 | 86.95 | 60.17 | 80.05 | 90.19 | 28.29 | 49.35 |
| | LDRE [63] | 17.96 | 18.32 | 20.21 | 21.11 | 25.69 | 55.13 | 69.04 | 89.90 | 60.53 | 80.65 | 90.70 | 24.81 | 45.63 |
| | *ImageScope** | 22.36 | 22.19 | 23.03 | 23.83 | 34.36 | 60.58 | 71.40 | 88.41 | 74.63 | 87.93 | 93.83 | 22.42 | 38.03 |
| | *ImageScope* | **25.26** | **25.82** | **27.15** | **28.11** | **38.43** | **66.27** | **76.96** | **91.83** | **75.93** | **89.21** | **94.63** | **31.42** | **50.80** |
| CLIP-ViT-L/14 | SEARLE [6] | 11.68 | 12.73 | 14.33 | 15.12 | 24.24 | 52.48 | 66.29 | 88.84 | 53.76 | 75.01 | 88.19 | 25.56 | 46.23 |
| | SEARLE-OTI [6] | 10.18 | 11.03 | 12.72 | 13.67 | 24.87 | 52.32 | 66.29 | 88.58 | 53.80 | 74.31 | 86.94 | 27.61 | 47.91 |
| | iSEARLE [1] | 12.50 | 13.61 | 15.36 | 16.25 | 25.28 | 54.00 | 66.72 | 88.80 | - | - | - | 27.52 | 48.96 |
| | iSEARLE-OTI [1] | 11.31 | 12.67 | 14.46 | 15.34 | 25.40 | 54.05 | 67.47 | 88.92 | - | - | - | 29.24 | 49.54 |
| | CIReVL [34] | 18.57 | 19.01 | 20.89 | 21.80 | 24.55 | 52.31 | 64.92 | 86.34 | 59.54 | 79.88 | 89.69 | 28.55 | 48.57 |
| | LDRE [63] | 23.35 | 24.03 | 26.44 | 27.50 | 26.53 | 55.57 | 67.54 | 88.50 | 60.43 | 80.31 | 89.90 | 28.51 | 50.54 |
| | HyCIR [63] | 18.91 | 19.67 | 21.58 | 22.49 | 25.08 | 53.49 | 67.03 | 89.85 | 53.83 | 75.06 | 87.18 | - | - |
| | LinCIR [23] | 12.59 | 13.58 | 15.00 | 15.85 | 25.04 | 53.25 | 66.68 | - | 57.11 | 77.37 | 88.89 | 26.28 | 46.48 |
| | FIT4CIR [41] | 15.05 | 16.32 | 18.06 | 19.05 | 25.90 | 55.61 | 67.66 | 89.66 | 55.21 | 75.88 | 87.98 | 29.42 | **50.88** |
| | *ImageScope** | 25.39 | 25.82 | 27.07 | 27.98 | 34.99 | 61.35 | 71.49 | 88.84 | 74.94 | 88.24 | 94.00 | 25.54 | 41.22 |
| | *ImageScope* | **28.36** | **29.23** | **30.81** | **31.88** | **39.37** | **67.54** | **78.05** | **92.94** | **76.36** | **89.40** | **95.21** | **31.36** | 50.78 |

methods exhibit significant improvements. The pre-aligned feature space of VLMs plays a crucial role in these methods and directly impacts the results.

• *ImageScope* compensates for the limitations of VLMs to some extent. Generally, it is unsurprising that smaller VLMs perform poorly. However, *ImageScope* shows strong performance even with smaller-scale CLIP-ViT-B/32. We give credit to the verification in Stage 2 and the evaluation in Stage 3, which refine the top retrieval results, thereby enhancing the retrieval accuracy.

**Text-to-Image Retrieval and Chat-based Image Retrieval.** Table 3 shows the comparison between the original VLMs and corresponding ones with *ImageScope*. We compare two versions of CLIP [31, 49] with different scales. We can observe consistent and significant improvements in different metrics across both datasets, indicating the superiority of our framework. Both the top-ranked R@1 and the overall ranking R@10 clearly outperform CLIP by a notable margin. This significant improvement is attributed to the verification in the second stage and the evaluation in the third stage, which together ensure that the retrieved results meet the requirements of the textual input. Figure 3 presents the comparison of Chat-IR on VisDial. Across various dialogue rounds, *ImageScope* consistently demonstrates superior retrieval performance, showing significant improvements over both CLIP and PlugIR. Additionally, CLIP's performance is constrained by the maximum length of its text input, resulting in subtle variations from the 7th round onward. Although PlugIR is capable of handling dialogue inputs, it remains suboptimal compared to our framework. The results of CIR, TIR, and Chat-IR demonstrate that *ImageScope* is capable of handling various LGIR tasks by accommodating different types of

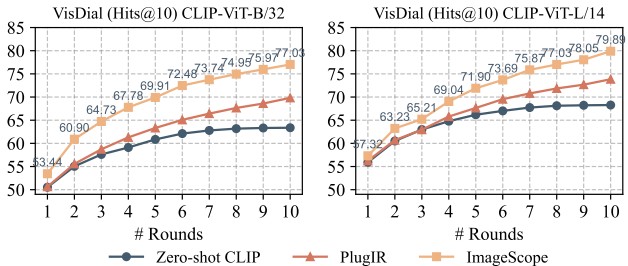

Figure 3: Performance of Chat-IR on VisDial [17] compared with Zero-shot CLIP [31] and PlugIR [37]. Complete results are shown in Table 6.

input and interaction forms, achieving effective performance in a training-free manner.

## 4.3 Ablation Study

**Stage Ablation.** To further investigate the impact of each designed stage of *ImageScope*, we conduct ablation study with on four LGIR datasets with different stages. "Stage1" means only including "Semantic Synthesis" stage, while "Stage2" means we add "Verification" after stage 1, and "Stage3" means we add "Evaluation" after Stage 2. As depicted in Figure 4, both Stage 2 "Verification" and Stage 3 "Evaluation" contribute to the improvement of top-retrieved results. We observe a significant improvement in the second stage compared to the first stage across different VLM scales. Moreover, despite only conducting pairwise evaluations on the top-3 candidate images in the third stage, the improvements in R@1 and H@1 are remarkable, especially on MSCOCO, CIRR, and VisDial datasets. This further validates the effectiveness of the evaluation stage design. These findings clearly highlight the critical role of both the verification

**Table 3: Performance comparison of TIR on Flickr30K and MSCOCO test sets. "∗" means using CLIP weights from [49].**

| Method | Flickr30K (1K test set) | | | MSCOCO (5K test set) | | | Average | | |
|---|---|---|---|---|---|---|---|---|---|
| | R@1 | R@5 | R@10 | R@1 | R@5 | R@10 | R@1 | R@5 | R@10 |
| CLIP-ViT-B/32* | 61.60 | 85.60 | 91.20 | 32.10 | 56.70 | 67.60 | 46.85 | 71.15 | 79.40 |
| *ImageScope** | 76.08 | 89.74 | 92.67 | 46.37 | 66.75 | 73.87 | 61.22 (+14.37) | 78.24 (+7.09) | 83.27 (+3.87) |
| CLIP-ViT-L/14* | 68.70 | 90.60 | 95.20 | 34.60 | 59.40 | 69.80 | 51.65 | 75.00 | 82.50 |
| *ImageScope** | 77.18 | 92.06 | 94.82 | 49.46 | 70.55 | 77.67 | 63.32 (+11.67) | 81.31 (+6.31) | 86.25 (+3.75) |
| CLIP-ViT-B/32 | 66.56 | 88.16 | 93.02 | 39.45 | 65.51 | 75.65 | 53.01 | 76.83 | 84.33 |
| *ImageScope* | 78.84 | 92.66 | 95.64 | 51.23 | 73.32 | 80.79 | 65.04 (+12.03) | 82.99 (+6.16) | 88.22 (+3.89) |
| CLIP-ViT-L/14 | 75.72 | 92.96 | 96.00 | 46.46 | 71.10 | 79.78 | 61.09 | 82.03 | 87.89 |
| *ImageScope* | 81.10 | 94.02 | 96.82 | 53.73 | 75.96 | 83.50 | 67.42 (+6.33) | 84.99 (+2.96) | 90.16 (+2.27) |

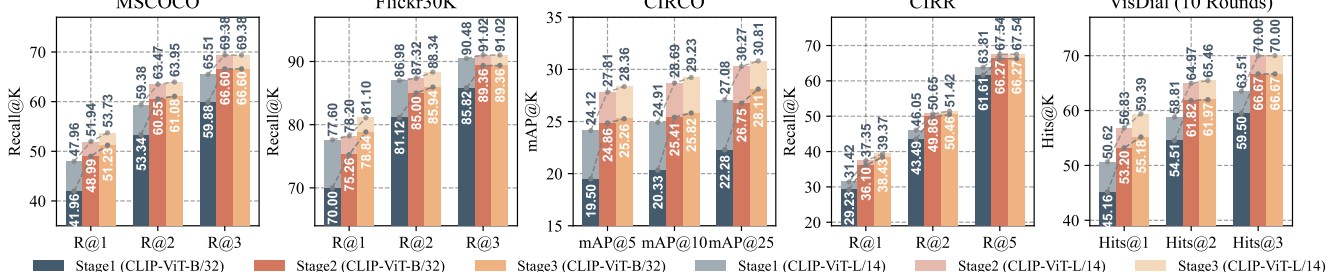

**Figure 4: Ablation study of each designed stage on five LGIR datasets. We show the results two scales of CLIP.**

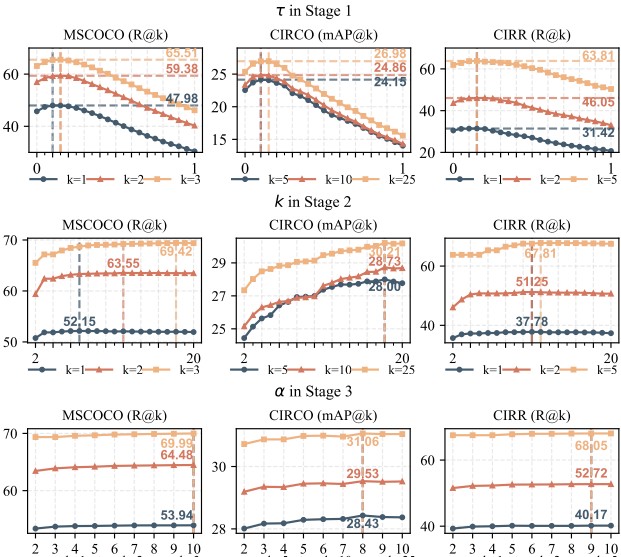

**Figure 5: Impact of parameter $\tau$ in Stage 1, $k$ in Stage 2 and $\alpha$ in Stage 3 on three datasets. We highlight the best metrics and corresponding values with numbers and dotted lines.**

and evaluation stages in enhancing performance and their pivotal impact on the final results.

**Impact of Parameters.** We take a further step and examine the impact of hyperparameters at each stage, *i.e.*, the weight $\tau$ in Stage 1, the number of candidate images $k$ in Stage 2 verification, and the number of paired evaluations $\alpha$ in Stage 3. As shown in Figure 5, we report the evaluation results of corresponding stages for a clear comparison. The first row of results regarding $\tau$ clearly shows a consistent trend of initial increase followed by a decline. Considering that when $\tau$ is set to 0, only text-to-image retrieval is performed,

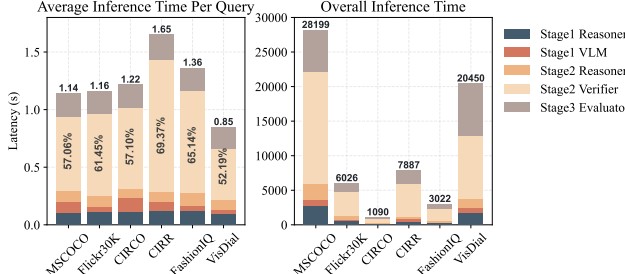

**Figure 6: Inference efficiency analysis. The left figure shows the average inference latency, and the right one shows the overall inference time. Numbers are shown in Tab. 7 and 8.**

this indicates that incorporating text-to-text retrieval helps improve performance. However, the value of $\tau$, representing the weight of text-to-text retrieval, should not be too large, as all datasets show that the optimal performance is achieved at 0.1 or 0.15. The results in the second and third rows represent the number of candidate images $k$ for verification and $\alpha$ for evaluation, respectively. Both exhibit an initial sharp improvement followed by a plateau, suggesting that incorporating more candidate images could enhance performance. These findings further confirm the effectiveness of each stage of the framework.

## 4.4 Discussion

*4.4.1 Efficiency Analysis.* Considering the use of LLMs and LMMs, we further explore the efficiency of *ImageScope* framework. Figure 6 illustrates the latency proportion at different stages for each query across various datasets, as well as the overall inference time. It can be observed that the inference time per query across all datasets is approximately 1 second. In the second stage, the verifier consumes over 50% of the time, with the CIRR dataset showing the highest proportion at 69.37%, as it requires more propositions to be verified

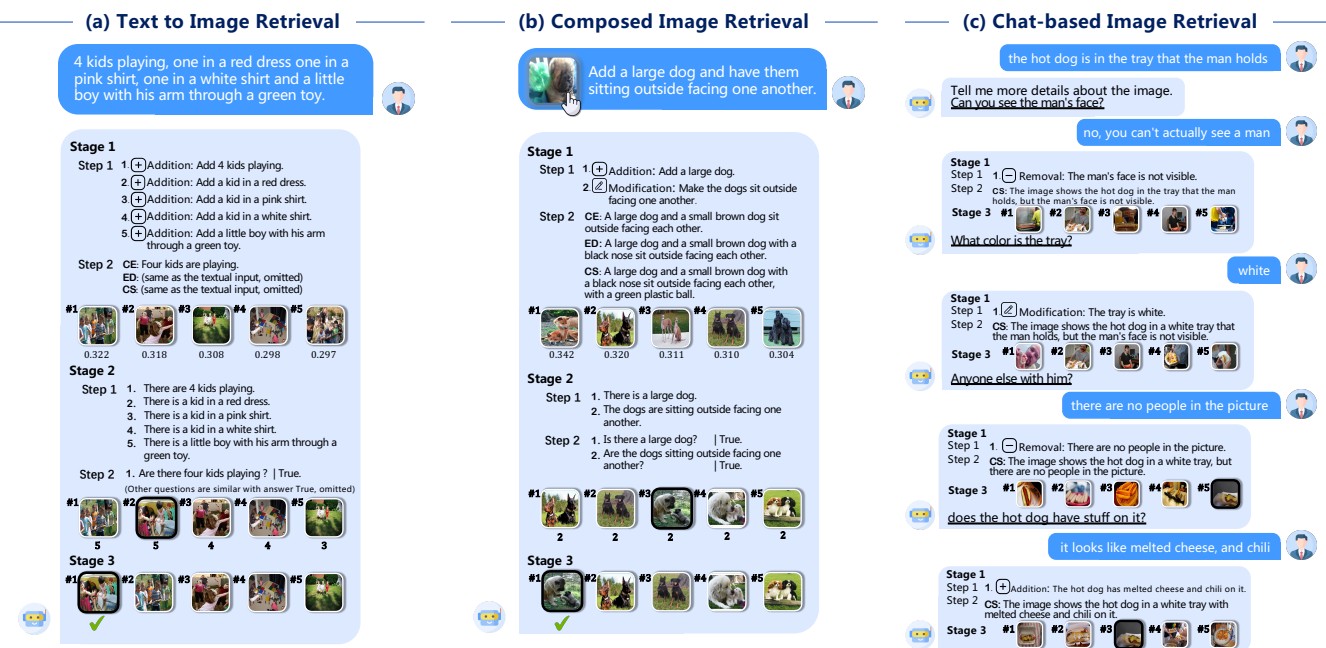

**Figure 7: Qualitative results. The underlined question in Chat-IR is from VisDial [17]. We show top-5 retrieved images and highlight the ground truth images with black borders.**

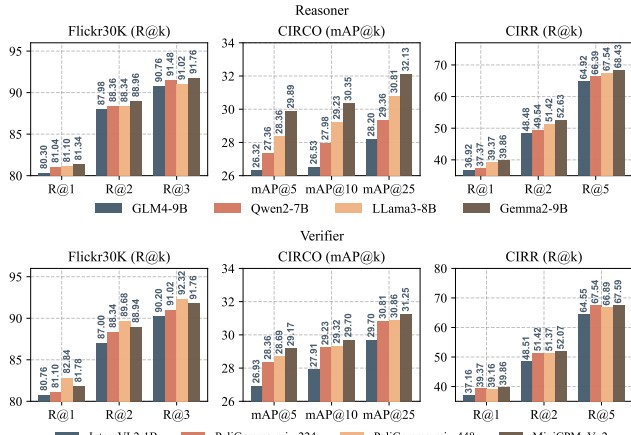

**Figure 8: Analysis of different reasoners and verifiers.**

for each query on average. Additionally, verifier perform verification on each proposition for $k$ candidate images individually, with $k$ set to 20. Therefore, considering the impact of $k$ as shown in Figure 5, reducing $k$ appropriately can provide a trade-off between performance and efficiency.

*4.4.2 Generality of LLM and LMM.* We conduct an investigation into the generality of the framework, particularly focusing on the crucial components, *i.e.*, the reasoner and verifier. As shown in Figure 8, we select various mainstream LLMs and LMMs. These results clearly demonstrate that *ImageScope* seamlessly integrates with different large models. Compared to the results of strong baselines in Tables 2 and 3, the results from various large models still show an advantage, further validating the generality and effectiveness of our framework. Moreover, it can be observed that more powerful LLMs (such as Gemma2) enhance reasoning, which in

turn improves retrieval performance. The results from the verifier indicate that increasing resolution (224 to 448) or model scale could also lead to further improvements in performance.

*4.4.3 Qualitative Results.* Finally, to more intuitively understanding the advantages of the proposed framework, we conduct an in-depth qualitative analysis. As shown in Figure 7, cases from various LGIR tasks are presented. In TIR task, *ImageScope* decomposes the user's input into a series of operations and propositions, successfully retrieving the correct image after the verification and evaluation stages. In the CIR task, *ImageScope* similarly reasons through feedback and retrieves images that largely meet the requirements. The evaluation in the third stage successfully retrieves the correct image, as evaluator performs pairwise comparison allows for better integration of reference images for reasoning. In Chat-IR task, it is evident that the user's intent has shifted, particularly regarding the presence of a "man." The qualitative analysis demonstrates that *ImageScope* can accurately understand the user's intent in multi-turn dialogues.

## 5 Conclusion

In this paper, we introduce *ImageScope*, a novel training-free framework designed to unify Language-Guided Image Retrieval (LGIR) tasks by harnessing the collective reasoning capabilities of large multimodal models. Additionally, to address the challenges posed by natural language ambiguity and complex image content, we propose a reflective method, termed verification-evaluation, for image retrieval. This method locally verifies predicate propositions and globally conducts pairwise evaluations. Experimental results on six widely-used LGIR datasets demonstrate the efficacy of the proposed framework. Ablation studies and comprehensive analysis underscore the generalizability of *ImageScope*.

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

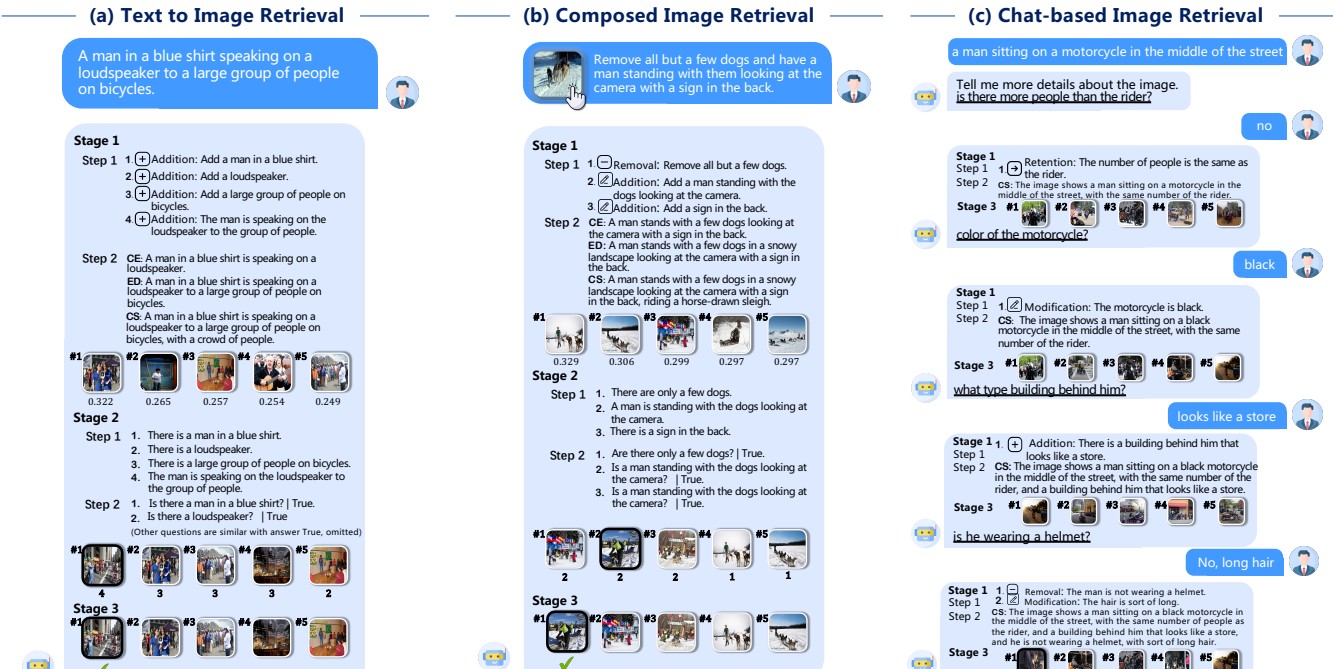

**Figure 9: More qualitative results. We show top-5 retrieved images and highlight the ground truth images with black borders.**

**Table 4: Performance comparison on CIRCO and CIRR test set. The best results are in bold, and the second best are underlined.** * means using CLIP weights from [49].

| VLM | Method | CIRCO | | | | CIRR | | | | | | |
|---|---|---|---|---|---|---|---|---|---|---|---|---|
| | | mAP@k | | | | Recall@k | | | | Recall$_{Subset}$@k | | |
| | | k=5 | k=10 | k=25 | k=50 | k=1 | k=5 | k=10 | k=50 | k=1 | k=2 | k=3 |
| CLIP-ViT-B/32 | PALAVRA [16] | 4.61 | 5.32 | 6.33 | 6.80 | 16.62 | 43.49 | 58.51 | 83.95 | 41.61 | 65.30 | 80.95 |
| | SEARLE [6] | 9.35 | 9.94 | 11.13 | 11.84 | 24.00 | 53.42 | 66.82 | 89.78 | 54.89 | 76.60 | 88.19 |
| | SEARLE-OTI [6] | 7.14 | 7.38 | 8.99 | 9.60 | 24.27 | 53.25 | 66.10 | 88.84 | 54.10 | 75.81 | 87.33 |
| | iSEARLE [1] | 10.58 | 11.24 | 12.51 | 13.26 | 25.23 | 55.69 | 68.05 | 90.82 | - | - | - |
| | iSEARLE-OTI [1] | 10.31 | 10.94 | 12.27 | 13.01 | 26.19 | 55.18 | 68.05 | 90.65 | - | - | - |
| | CIReVL [34] | 14.94 | 15.42 | 17.00 | 17.82 | 23.94 | 52.51 | 66.00 | 86.95 | 60.17 | 80.05 | 90.19 |
| | LDRE [63] | 17.96 | 18.32 | 20.21 | 21.11 | 25.69 | 55.13 | 69.04 | 89.90 | 60.53 | 80.65 | 90.70 |
| | *ImageScope** | 22.36 | 22.19 | 23.03 | 23.83 | 34.36 | 60.58 | 71.40 | 88.41 | 74.63 | 87.93 | 93.83 |
| | *ImageScope* | **25.26** | **25.82** | **27.15** | **28.11** | **38.43** | **66.27** | **76.96** | **91.83** | **75.93** | **89.21** | **94.63** |
| CLIP-ViT-L/14 | Pic2Word [51] | 8.72 | 9.51 | 10.64 | 11.29 | 23.90 | 51.70 | 65.30 | 87.80 | - | - | - |
| | SEARLE [6] | 11.68 | 12.73 | 14.33 | 15.12 | 24.24 | 52.48 | 66.29 | 88.84 | 53.76 | 75.01 | 88.19 |
| | SEARLE-OTI [6] | 10.18 | 11.03 | 12.72 | 13.67 | 24.87 | 52.32 | 66.29 | 88.58 | 53.80 | 74.31 | 86.94 |
| | iSEARLE [1] | 12.50 | 13.61 | 15.36 | 16.25 | 25.28 | 54.00 | 66.72 | 88.80 | - | - | - |
| | iSEARLE-OTI [1] | 11.31 | 12.67 | 14.46 | 15.34 | 25.40 | 54.05 | 67.47 | 88.92 | - | - | - |
| | CIReVL [34] | 18.57 | 19.01 | 20.89 | 21.80 | 24.55 | 52.31 | 64.92 | 86.34 | 59.54 | 79.88 | 89.69 |
| | LDRE [63] | 23.35 | 24.03 | 26.44 | 27.50 | 26.53 | 55.57 | 67.54 | 88.50 | 60.43 | 80.31 | 89.90 |
| | HyCIR [32] | 18.91 | 19.67 | 21.58 | 22.49 | 25.08 | 53.49 | 67.03 | 89.85 | 53.83 | 75.06 | 87.18 |
| | LinCIR [23] | 12.59 | 13.58 | 15.00 | 15.85 | 25.04 | 53.25 | 66.68 | - | 57.11 | 77.37 | 88.89 |
| | FIT4CIR [41] | 15.05 | 16.32 | 18.06 | 19.05 | 25.90 | 55.61 | 67.66 | 89.66 | 55.21 | 75.88 | 87.98 |
| | *ImageScope** | 25.39 | 25.82 | 27.07 | 27.98 | 34.99 | 61.35 | 71.49 | 88.84 | 74.94 | 88.24 | 94.00 |
| | *ImageScope* | **28.36** | **29.23** | **30.81** | **31.88** | **39.37** | **67.54** | **78.05** | **92.94** | **76.36** | **89.40** | **95.21** |

---

**Prompt1: Stage 1 Reasoner**

# Task Description
You are given a description of Image Retrieval. The task is to combine information from both textual instruction and reference image or information to accurately retrieve images. You need to follow two steps to derive "what does the target image look like".

## Step 1: Instruction Classification and Impact Analysis
Classify the given instruction into the following types and identify how it affects the reference image. For each type, determine the specific elements or attributes of the reference image that are impacted. The instruction types are:
    (1) Addition: Introduces new elements or features to the reference image. Identify which existing element the addition relates to or where it should be placed.
    (2) Removal: Eliminates certain elements from the reference image. Identify which existing element is removed.
    (3) Modification: Alters attributes of existing elements in the reference image. Determine which specific element is being modified and how.
    (4) Comparison: Contrasts elements in the reference image using terms like "different," "same," "more," or "less.". Identify elements and attributes being compared.
    (5) Retention: Specifies certain existing elements in the reference image to remain unchanged. Ensure these elements are noted for inclusion in the target image.

## Step 2: Target Image Description
Describe what the target image should look like based on the instruction and reference image analysis. Provide three sentences, each focusing on a different semantic aspect:
    (1) Core Elements: Mention only the elements that appear in the instruction without necessary adjectives.
    (2) Enhanced Details: Mention the elements in the instruction with necessary adjectives from the reference image.
    (3) Comprehensive Synthesis: Mention both the elements in the instruction and relevant elements in the reference image with necessary adjectives.

The instruction and reference image description will be given to you to solve the task. Refer to the following examples and the final output should in JSON format.
—
Here is an example:
### Query
   -   Instruction: has the person holding a baby
   -   Reference Image: A woman with dark hair is smiling under a gray umbrella with a white flower hanging from it.

### Solve
1. **Step 1.** Based on the instruction:
   -   Addition: Make the woman holding a baby.

2. **Step 2.** Based on step 1, the target image should be like:
   -   A woman holds a baby.
   -   A woman with dark hair holds a baby under an umbrella.
   -   A woman with dark hair holds a baby and is smiling, under a gray umbrella.
—
(In-context examples)
—
Below is the query you need to solve:
### Query
   -   Instruction: [[INSTRUCTION]]
   -   Reference Image: [[REF_IMAGE_DESC]]

**Figure 10: The prompt we use for *reasoner* in the first stage. [[INSTRUCTION]] and [[REF_IMAGE_DESC]] are placeholders that can be replaced by a input query.**

---

**Prompt2: Stage 2 Reasoner**

# Task Description
The task of Atomic Proposition Generation involves breaking down a instruction into multiple simple, verifiable propositions, each having a unique answer that is either True (Yes) or False (No). Based on the provided instruction and a target image description, you need to break down the instruction into several atomic propositions and corresponding answers, following the two steps below.

## Step 1: Statement Sentence Conversion
Convert each atomic instruction into statement sentence. There are five types of atomic instruction: addition, removal, modification, comparison and retention.

## Step 2: Question Form Conversion
Convert each statement sentence into questions, also provide the ground truth answer based on the given instruction.

The instruction and atomic instructions will be given to you to solve the task.
—
Here is an example:

### Query
- Instruction: has the person holding a baby
- Atomic Instructions:
  (1) Addition: Make the woman holding a baby.

### Solve
1. **Step 1.** Based on the atomic instructions, the statements are:
   (1) There is a woman holding a baby.

2. **Step 2.** Based on step 1, the questions and answers are:
   (1) Q: Is there a woman holding a baby? A: Yes. (True)
—
(In-context examples)
—
Below is the query you need to solve:

### Query
- Instruction: [[INSTRUCTION]]
- Atomic Instructions: [[ATOMIC_INST]]

---

**Figure 11: The prompt we use for *reasoner* in the second stage. [[INSTRUCTION]] and [[ATOMIC_INST]] are placeholders. [[INSTRUCTION]] is replaced by language feedback of a query, and [[ATOMIC_INST]] is replaced by the output from step 1 of the first stage.**

---

**Prompt3: Stage 3 Evaluator**

Your task is to evaluate and determine if the right candidate image reflects the change described in the <INSTRUCTION> "[[INSTRUCTION]]". The instruction may describe:
1. A change from the left reference image to the right candidate image, or
2. The direct desired appearance of the right candidate image itself.

**Steps:**
1. For change-based instructions:
   a. Analyze the left reference image as the starting point.
   b. Examine the right candidate image for the described change.
2. For direct description instructions:
   a. Focus solely on the right candidate image.
   b. Determine if it matches the instruction's description.
3. Provide your answer as follows:
   ANSWER: [Yes/No]
   Where:
   - 'Yes' if the candidate image correctly matches the <INSTRUCTION>.
   - 'No' if it fails to match the <INSTRUCTION> .
4. After the ANSWER line, briefly explain how the candidate image does or does not match the <INSTRUCTION>.

**Important notes:**
   - Base your analysis SOLELY on the <INSTRUCTION> and relevant image(s).
   - Ignore elements irrelevant to the <INSTRUCTION> .
   - Do not introduce criteria beyond the <INSTRUCTION> .

Always start with the ANSWER line, followed by your explanation on a new line.

---

**Figure 12: The prompt we use for *evaluator* in the third stage. [[INSTRUCTION]] is a placeholder, which is replaced by language feedback of a query.**

**Table 5: Performance comparison on FashionIQ validation set. The best results are in bold, and the second best are underlined. \* means using CLIP weights from [49].**

| VLM | Method | Shirt | | Dress | | Toptee | | *Avg.* | |
|---|---|---|---|---|---|---|---|---|---|
| | | R@10 | R@50 | R@10 | R@50 | R@10 | R@50 | R@10 | R@50 |
| CLIP-ViT-B/32 | PALAVRA [16] | 21.49 | 37.05 | 17.25 | 35.94 | 20.55 | 38.76 | 19.76 | 37.25 |
| | SEARLE [6] | 24.44 | 41.61 | 18.54 | 39.51 | 25.70 | 46.46 | 22.89 | 42.53 |
| | SEARLE-OTI [6] | 25.37 | 41.32 | 17.85 | 39.91 | 24.12 | 45.79 | 22.45 | 42.34 |
| | iSEARLE [1] | 25.81 | 43.52 | 20.92 | 42.19 | 26.47 | 48.70 | 24.40 | 44.80 |
| | iSEARLE-OTI [1] | 27.09 | 43.42 | 21.27 | 42.19 | 26.82 | 48.75 | 25.06 | 44.79 |
| | CIReVL [34] | 28.36 | 47.84 | 25.29 | **46.36** | 31.21 | 53.85 | 28.29 | 49.35 |
| | LDRE [63] | 27.38 | 46.27 | 19.97 | 41.84 | 27.07 | 48.78 | 24.81 | 45.63 |
| | *ImageScope*\* | 24.29 | 37.49 | 18.00 | 35.20 | 24.99 | 41.41 | 22.42 | 38.03 |
| | *ImageScope* | **31.65** | **50.15** | **26.82** | 46.31 | **35.80** | **55.94** | **31.42** | **50.80** |
| CLIP-ViT-L/14 | Pic2Word [51] | 26.20 | 43.60 | 20.00 | 40.20 | 27.90 | 47.40 | 24.70 | 43.73 |
| | SEARLE [6] | 26.89 | 45.58 | 20.48 | 43.13 | 29.32 | 49.97 | 25.56 | 46.23 |
| | SEARLE-OTI [6] | 30.37 | 47.49 | 21.57 | 44.47 | 30.90 | 51.76 | 27.61 | 47.91 |
| | iSEARLE [1] | 28.75 | 47.84 | 22.51 | 46.36 | 31.31 | 52.68 | 27.52 | 48.96 |
| | iSEARLE-OTI [1] | 31.80 | 50.20 | 24.19 | 45.12 | 31.72 | 53.29 | 29.24 | 49.54 |
| | CIReVL [34] | 29.49 | 47.40 | 24.79 | 44.76 | 31.36 | 53.65 | 28.55 | 48.57 |
| | LDRE [63] | 31.04 | **51.22** | 22.93 | 46.76 | 31.57 | 53.64 | 28.51 | 50.54 |
| | LinCIR [23] | 29.10 | 46.81 | 20.92 | 42.44 | 28.81 | 50.18 | 26.28 | 46.48 |
| | FTI4CIR [41] | 31.35 | 50.59 | 24.49 | **47.84** | 32.43 | 54.21 | 29.42 | **50.88** |
| | *ImageScope*\* | 27.82 | 41.76 | 20.18 | 37.48 | 28.61 | 44.42 | 25.54 | 41.22 |
| | *ImageScope* | **32.87** | 51.07 | **26.17** | 46.15 | **35.03** | **55.12** | **31.36** | 50.78 |

**Table 6: Performance comparison on VisDial validation set. We re-implement PlugIR [37] with LLaMA3-8B [2] and CLIP [31] for a fair comparison. We report H@1 and H@10 in the following table.**

| VLM | Method | VisDial #Round (Hits@1) | | | | | | | | | |
|---|---|---|---|---|---|---|---|---|---|---|---|
| | | 1 | 2 | 3 | 4 | 5 | 6 | 7 | 8 | 9 | 10 |
| CLIP-ViT-B/32 | Zero-shot CLIP [31] | 22.53 | 26.55 | 28.73 | 29.84 | 31.49 | 32.41 | 33.33 | 33.62 | 33.72 | 33.77 |
| | PlugIR [37] | 22.75 | 25.55 | 27.70 | 30.72 | 32.80 | 34.54 | 36.05 | 37.55 | 38.37 | 39.49 |
| | *ImageScope* | 22.67 | 31.54 | 36.72 | 40.50 | 44.04 | 47.53 | 49.76 | 51.45 | 53.54 | 55.18 |
| CLIP-ViT-L/14 | Zero-shot CLIP [31] | 29.51 | 33.14 | 35.32 | 36.87 | 38.13 | 39.24 | 40.16 | 40.36 | 40.60 | 40.60 |
| | PlugIR [37] | 29.53 | 33.62 | 35.90 | 39.24 | 41.28 | 43.07 | 44.33 | 45.16 | 45.98 | 47.24 |
| | *ImageScope* | 26.74 | 35.80 | 42.10 | 47.04 | 49.66 | 52.71 | 55.14 | 56.49 | 58.28 | 59.40 |

| VLM | Method | VisDial #Round (Hits@10) | | | | | | | | | |
|---|---|---|---|---|---|---|---|---|---|---|---|
| | | 1 | 2 | 3 | 4 | 5 | 6 | 7 | 8 | 9 | 10 |
| CLIP-ViT-B/32 | Zero-shot CLIP [31] | 50.53 | 55.04 | 57.61 | 59.11 | 60.85 | 62.11 | 62.79 | 63.18 | 63.32 | 63.37 |
| | PlugIR [37] | 50.64 | 55.57 | 58.72 | 61.29 | 63.32 | 65.07 | 66.42 | 67.64 | 68.60 | 69.82 |
| | *ImageScope* | 53.44 | 60.90 | 64.73 | 67.78 | 69.91 | 72.48 | 73.74 | 74.95 | 75.97 | 77.03 |
| CLIP-ViT-L/14 | Zero-shot CLIP [31] | 55.91 | 60.51 | 62.98 | 64.78 | 66.18 | 67.01 | 67.73 | 68.12 | 68.22 | 68.27 |
| | PlugIR [37] | 56.23 | 60.69 | 62.94 | 65.79 | 67.64 | 69.53 | 70.78 | 71.85 | 72.67 | 73.84 |
| | *ImageScope* | 57.32 | 63.23 | 65.21 | 69.04 | 71.90 | 73.69 | 75.87 | 77.03 | 78.05 | 79.89 |

**Table 7: Numerical results of average inference latency (second) per query on LGIR datasets.**

| Stage | MSCOCO | Flickr30K | CIRCO | CIRR | F-Dress | F-Shirt | F-Toptee | FashionIQ *Avg.* | VisDial |
|---|---|---|---|---|---|---|---|---|---|
| Stage1 Reasoner | 0.109 | 0.117 | 0.114 | 0.122 | 0.139 | 0.111 | 0.113 | 0.121 | 0.097 |
| Stage1 VLM | 0.091 | 0.035 | 0.118 | 0.075 | 0.046 | 0.045 | 0.045 | 0.045 | 0.035 |
| Stage2 Reasoner | 0.091 | 0.103 | 0.086 | 0.089 | 0.110 | 0.108 | 0.113 | 0.110 | 0.086 |
| Stage2 Verifier | 0.651 | 0.715 | 0.695 | 1.146 | 0.929 | 0.837 | 0.898 | 0.889 | 0.444 |
| Stage3 Evaluator | 0.200 | 0.192 | 0.204 | 0.220 | 0.196 | 0.198 | 0.202 | 0.199 | 0.187 |
| Total Latency | 1.141 | 1.163 | 1.217 | 1.652 | 1.420 | 1.299 | 1.371 | 1.364 | 0.850 |

**Table 8: Numerical results of overall inference time (second) on LGIR datasets.**

| Stage | MSCOCO | Flickr30K | CIRCO | CIRR | F-Dress | F-Shirt | F-Toptee | FashionIQ *Avg.* | VisDial |
|---|---|---|---|---|---|---|---|---|---|
| Stage1 Reasoner | 2728 | 587 | 92 | 505 | 280 | 226 | 221 | 243 | 1724 |
| Stage1 VLM | 907 | 175 | 95 | 311 | 92 | 93 | 89 | 91 | 718 |
| Stage2 Reasoner | 2263 | 516 | 68 | 369 | 223 | 220 | 221 | 221 | 1360 |
| Stage2 Verifier | 16291 | 3571 | 557 | 4803 | 1879 | 1703 | 1757 | 1780 | 9057 |
| Stage3 Evaluator | 6010 | 1177 | 278 | 1899 | 730 | 652 | 678 | 687 | 7591 |
| Total Time | 28199 | 6027 | 1089 | 7888 | 3204 | 2894 | 2967 | 3022 | 20450 |

