# OpenReview forum: "ImageScope: Unifying Language-Guided Image Retrieval via Large Multimodal Model Collective Reasoning"
_ACM.org/TheWebConf/2025/Conference — WWW 2025 Oral_

### Official Review · Reviewer_CVkx · 2024-11-24

**Novelty:** 6
**Technical Quality:** 6

**Review:**

The quality and originality of the work are good, and it adopts the big language model of the thought chain and the big multimodal model, which is of great interest to the community.

**Questions:**

Can the code be published to verify the authenticity of the experimental results?

**Reviewer Confidence:**

3: The reviewer is confident but not certain that the evaluation is correct

**Scope:**

4: The work is relevant to the Web and to the track, and is of broad interest to the community

---

### Official Review · Reviewer_8JT8 · 2024-11-27

**Novelty:** 4
**Technical Quality:** 3

**Review:**

The paper highlights the challenge of addressing variants of language guided image retrieval (LGIR) tasks through separate systems, which increases the system complexity and maintenance costs. The significance of developing a unified systems for various LGIR tasks are therefore clearly stated by authors and the solution to a unified framework is worth to be researched. To bridge this gap, this paper proposes a prompt-based three-stage framework with pre-trained vision large language models. The effectiveness of the proposed system is validated through experiments.

Pros:
1. The research question is well-defined and addresses a significant issue in LGIR. The focus on a unified framework for LGIR tasks and the innovative use of LMMs are meaningful and less explored on this topic.
2. The proposed framework adopts an innovative verification-evaluation method to enhance the system performance, which proved to be useful in LGIR task.
3. The writing of this paper is overall easy to follow, except for some parts need further clarifications.

Cons:
1. The proposed framework integrates five (large) pretrained models (CLIP, LLaVA, LLaMA3, PaliGemma, and InternVL) with a total size exceeding 25 billion parameters. This raises concerns about the framework’s complexity and cost, potentially undermining the main goal of simplifying LGIR systems.
2. While the framework achieves a Recall@10 of 78.05 on the CIRR dataset, this improvement over smaller models like CIReVL (~2B with ViT-G parameters, Recall@10 = 75.06) is modest. It is unclear whether this gain is obtained from the innovative framework design or merely from the increased size of pre-trained models.
3. Some sections of the paper are not clearly written for me, which may hinder readers' understanding of key points. (please see questions)

Summary:
The paper addresses a meaningful research problem, offering a novel approach to unify various LGIR task systems. However, concerns regarding model complexity, incremental performance gains, and writing clarity are suggested be addressed to strengthen the work. Clarifying the framework's scalability and justifying its benefits over smaller, less resource-intensive models could further enhance its impact.

**Questions:**

1. Your framework integrates five large models with a total size about 25B parameters. How does this address the issue of system complexity and cost, which you identified as a key challenge in LGIR tasks?
2. If smaller model such as CIReVl can achieve a comparable zero-shot performance (as we mentioned in review summary section), can you please justify the significant of gains bought by the innovative framework design?
3. The writing tries to differentiate VLM and LMM terms. However, the difference between them are still not very clear. Can you further clarify them?
4. The "atomic instruction" seems an important section for the proposed model. But I am still not clear about how to obtain the instruction?

(If the above questions are properly addressed, I am more than happy to change my rating to higher level :))

**Reviewer Confidence:**

4: The reviewer is certain that the evaluation is correct and very familiar with the relevant literature

**Scope:**

4: The work is relevant to the Web and to the track, and is of broad interest to the community

---

### Official Review · Reviewer_zD7a · 2024-11-30

**Novelty:** 5
**Technical Quality:** 4

**Review:**

**Summary:**

This paper introduces ImageScope, a training-free framework for language-guided image retrieval (LGIR) that unifies tasks like text-to-image retrieval, composed image retrieval, and chat-based image retrieval within a single system. This unified strategy aims to streamline LGIR tasks, reducing system complexity while enhancing retrieval accuracy and robustness.

**Strengths:**
1. The paper is clearly written, with a well-organized structure that facilitates understanding of the framework and its contributions.
2. The proposed framework unifies three tasks and training free.
3. Experiments show the effectiveness of the proposed method.

**Questions:**

1. Further clarification is needed in the description of the CoT-based semantic synthesis process, particularly regarding how the framework handles complex or ambiguous instructions in practice.

2. The current ablation study of the Impact of Parameters could be more detailed.

3. More complex case studies and visual qualitative results interpretation and analysis will be more helpful to reflect the effectiveness of the paper method.

4. Why in Implementation Details the weight 𝜏 in stage 1 is set to 0.15. The number of candidate images to verify in stage 2, i.e.,  𝑘 is set to 20. The number of images to evaluate in stage 3 𝛼 is set to 3.
But **4.3 Ablation Impact of Parameters** in the study the number of images to evaluate in stage 3 looking at alpha the best result is between 8 and 10, The number of candidate images to verify in stage 2 and 20 is not the best, it seems that the best value of k is different from task to task, how do you get uniform here

**Reviewer Confidence:**

3: The reviewer is confident but not certain that the evaluation is correct

**Scope:**

4: The work is relevant to the Web and to the track, and is of broad interest to the community

---

### Official Review · Reviewer_k2fy · 2024-12-02

**Novelty:** 7
**Technical Quality:** 7

**Review:**

The paper proposes a unified framework based on large multimodal models for text-guided image retrieval. The motivation for the proposed method is clear, with significant originality, and its effectiveness is demonstrated through extensive analytical experiments. The specific advantages are as follows:

1. Using text as an intermediate modality fully leverages the semantic understanding and reasoning capabilities of existing large language models. This is a very straightforward and effective insight, unifying three different text-guided image retrieval problems under the same paradigm without requiring training, which presents significant application potential.
2. The experiments and analysis are thorough, with experiments conducted on multiple datasets and using various backbone models, fully validating the method's generalizability. Ablation experiments sufficiently confirm the rationality of the designed modules.

To the best of my knowledge, I do not perceive any obvious shortcomings in this paper, and thus I give it a full score.

**Questions:**

To the best of my knowledge, I do not perceive any obvious shortcomings in this paper.

**Reviewer Confidence:**

3: The reviewer is confident but not certain that the evaluation is correct

**Scope:**

4: The work is relevant to the Web and to the track, and is of broad interest to the community